# Finite-horizon Equilibria for Neuro-symbolic Concurrent Stochastic Games

Rui Yan [*1]    Gabriel Santos [*1]    Xiaoming Duan[2]    David Parker[3]    Marta Kwiatkowska[1]

[1]Department of Computer Science, University of Oxford, Oxford, UK
[2]Department of Automation, Shanghai Jiao Tong University, Shanghai, China
[3]School of Computer Science, University of Birmingham, Birmingham, UK

## Abstract

We present novel techniques for neuro-symbolic concurrent stochastic games, a recently proposed modelling formalism to represent a set of probabilistic agents operating in a continuous-space environment using a combination of neural network based perception mechanisms and traditional symbolic methods. To date, only zero-sum variants of the model were studied, which is too restrictive when agents have distinct objectives. We formalise notions of equilibria for these models and present algorithms to synthesise them. Focusing on the finite-horizon setting, and (global) social welfare subgame-perfect optimality, we consider two distinct types: Nash equilibria and correlated equilibria. We first show that an exact solution based on backward induction may yield arbitrarily bad equilibria. We then propose an approximation algorithm called frozen subgame improvement, which proceeds through iterative solution of nonlinear programs. We develop a prototype implementation and demonstrate the benefits of our approach on two case studies: an automated car-parking system and an aircraft collision avoidance system.

## 1 INTRODUCTION

Stochastic games [Shapley, 1953] are a well established model for the formal design and analysis of probabilistic multi-agent systems. In particular, *concurrent stochastic games* (CSGs) provide a natural framework for modelling a set of interactive, rational agents operating concurrently within an uncertain or probabilistic environment. For finite-state CSGs, algorithms for their solution are known [Chatterjee et al., 2013, de Alfaro and Majumdar, 2004, De Alfaro et al., 2007] and, more recently, techniques and tools for

their formal modelling, analysis and verification have been developed [Kwiatkowska et al., 2020, 2021] and applied to examples across robotics, computer security and networks.

In more complex scenarios, for example sequential decision making in continuous-state or mixed discrete-continuous state environments, CSGs are again a natural formalism for problems such as multi-agent reinforcement learning [Papoudakis et al., 2021, Yan et al., 2022a]. A recent trend in this setting is the use of neural networks (NNs), to represent learnt approximations to value functions [Omidshafiei et al., 2017] or strategies [Lowe et al., 2017] for CSGs. However, the scalability and efficiency of such approaches are limited when NNs are used to manage multiple, complex aspects of the system. To overcome this, a further promising direction is the use of *neuro-symbolic* approaches. These deploy NNs within certain data-driven components of the control problem, e.g., for perception modules, and traditional symbolic methods for others, e.g., nonlinear controllers.

In this paper, we work with the recently proposed formalism of *neuro-symbolic concurrent stochastic games* (NS-CSGs) [Yan et al., 2022b], designed to model probabilistic multi-agent systems comprising neuro-symbolic agents operating concurrently within a shared, continuous-state environment. In [Yan et al., 2022b], the *zero-sum* control problem is considered, namely to synthesise strategies for one set of agents who are aiming to maximise their (discounted, infinite-horizon) expected reward, while the other agents aim to minimise this value. However, in practice, this is limiting: even for the case of just two coalitions of agents, they will often have distinct, but not directly opposing goals, which cannot be modelled in a zero-sum fashion.

To tackle this problem, we work with *equilibria*, defined by a separate, independent objective for each agent. These are particularly attractive since they ensure stability against deviations by individual agents, improving the overall system outcomes. We formalise the equilibrium synthesis problem for NS-CSGs, considering two distinct variants: *Nash equilibria* (NEs), which aim to ensure that no agent has

---
[*]Equal Contributions.

*Accepted for the 38th Conference on Uncertainty in Artificial Intelligence* (UAI 2022).

an incentive to deviate unilaterally from their strategy, and *correlated equilibria* (CEs), which allow agent coordination, e.g., through public signals, and where agents have no incentive to deviate from the resulting actions. The latter can both simplify strategy synthesis and improve performance.

Our focus is on (undiscounted) *finite-horizon* objectives, which simplifies the analysis (note that the existence of infinite-horizon NE for CSGs is an open problem [Bouyer et al., 2014], and the verification of non-probabilistic infinite-horizon reachability properties for neuro-symbolic games is undecidable [Akintunde et al., 2020a]), but also has a number of useful applications, e.g. in receding horizon control. Since multiple equilibria may exist, we target *social welfare (SW)* optimal equilibria, which maximise the sum of the individual agent objectives.

We also work with *subgame-perfect equilibria* (SPE), which are equilibria in every state of the game, ensuring that optimality remains as later states of the game are reached [Abreu et al., 2020, Fudenberg and Levine, 2009, Littman et al., 2006, Osborne et al., 2004]. Crucially, we consider *globally optimal* equilibria which, from a fixed initial state, are optimal over the chosen time horizon. This is in contrast to techniques for equilibria in finite-state CSGs [Kwiatkowska et al., 2021, 2022], which consider only local optimality at each time step in the finite-horizon setting.

We first adapt (classical) backward induction to NS-CSGs based on local optimality, but show that it may find an arbitrarily bad SPE. Then, for a fixed initial state, we show how to compute optimal equilibria by unfolding the game tree (including invocation of the NN perception function) and solving a *nonlinear program*. However, this suffers from limited scalability. So we then propose *frozen subgame improvement* (FSI), an approximation algorithm which iteratively solves nonlinear programs to monotonically improve the social welfare. Our approach is wholly different from the zero-sum (discounted, infinite-horizon) solution of NS-CSGs in [Yan et al., 2022b], which applies value/policy iteration to finite model abstractions that rely on assumptions about the functions used to specify the model.

Finally, we implement our algorithms and evaluate them on two case studies, a car-parking example and the VerticalCAS (VCAS) aircraft system for collision avoidance, showing that they are capable of automatically generating equilibria that can improve over zero-sum strategies.

**Related Work.** Several papers have considered verification and synthesis of equilibria for stochastic games [Fernando et al., 2018, Horák and Bošanský, 2019, Kwiatkowska et al., 2021, Mari et al., 2009], aiming to prove that a game satisfies a given equilibrium-related requirement specification and also to find such an equilibrium. However, none of these support CSGs whose agents are partly realized via NNs. The PRISM-games tool [Kwiatkowska et al., 2020] provides modelling, verification and equilibria synthesis for (discrete-state) CSGs, including finite-horizon analysis via backward induction, but for the simpler case of local optimality, as discussed abvove. [Kwiatkowska et al., 2020] also includes infinite-horizon $\epsilon$-optimal social welfare Nash equilibria, and [Kwiatkowska et al., 2022] correlated equilibria with two types of optimality conditions, computed using value iteration, but again only for discrete models.

Numerous methods have been proposed to compute SPEs since their introduction in the 1970s [Selten, 1975]. Most of these address the infinite horizon, for which fixed-point algorithms are the most common methods, from operator design for SPE payoff correspondence [Abreu et al., 2020, Brihaye et al., 2020, Burkov and Chaib-draa, 2010, Kitti, 2016, Yeltekin et al., 2017], to homotopy methods [Li and Dang, 2020]. For the finite horizon, which we consider here for reasons of decidability, backward induction is a simple and common bottom-up algorithm for finding an SPE efficiently. However, all these approaches fail to identify SW-SPEs over a finite horizon. In [Littman et al., 2006], a polynomial algorithm is proposed for computing optimal SPEs for turn-based games played over trees, which cannot deal with the concurrency in CSGs.

Neuro-symbolic computing has been attracting attention recently, see [Kahneman, 2011] and the surveys [De Raedt et al., 2020, Lamb et al., 2020]. The works of [Akintunde et al., 2020a,b] consider neuro-symbolic multi-agent systems represented as neural interpreted systems and study the finite-horizon verification problem for Alternating Temporal Logic, solved through reduction to an MILP problem, but no equilibria properties. The agents are endowed with perception similarly to what we do here, but are not stochastic.

## 2 NEURO-SYMBOLIC CSGS

We begin by describing *neuro-symbolic concurrent stochastic games (NS-CSGs)* [Yan et al., 2022b], the modelling formalism that we use in this paper, for which we then define our notions of equilibria.

An NS-CSG comprises a number of interacting neuro-symbolic agents acting in a shared environment. Each agent has finitely many local states and actions, and is additionally endowed with a perception mechanism implemented as a neural network (NN), through which it can observe the state of the environment, storing the observations locally in *percepts*. For the purposes of this paper it suffices to assume that an NN is a function $f : \mathbb{R}^{m_1} \to \mathbb{R}^{m_2}$ over finite real vector spaces. Formally, an NS-CSG is defined as follows.

**Definition 1** (NS-CSG). *A neuro-symbolic concurrent stochastic game (NS-CSG)* $\mathsf{C}$ *comprises agents* $(\mathsf{Ag}_i)_{i \in N}$, *for* $N = \{1, \ldots, n\}$, *and an environment* $E$ *where:*

$$\mathsf{Ag}_i = (S_i, A_i, \Delta_i, obs_i, \delta_i) \text{ for } i \in N, \quad E = (S_E, \delta_E)$$

*and we have:*

- $S_i = Loc_i \times Per_i$ *is a set of states for* $\mathsf{Ag}_i$, *where* $Loc_i \subseteq \mathbb{R}^{b_i}$ *and* $Per_i \subseteq \mathbb{R}^{d_i}$ *are finite sets of local states and percepts, respectively;*

- $S_E \subseteq \mathbb{R}^e$ *is a finite or infinite set of environment states;*

- $A_i$ *is a nonempty finite action set for* $\mathsf{Ag}_i$, *and* $A := (A_1 \cup \{\perp\}) \times \cdots \times (A_n \cup \{\perp\})$ *is the set of joint actions, where* $\perp$ *is an idle action disjoint from* $\cup_{i=1}^n A_i$;

- $\Delta_i : S_i \to 2^{A_i}$ *is an available action function, defining the actions* $\mathsf{Ag}_i$ *can take in each state;*

- $obs_i : (S_1 \times \cdots \times S_n \times S_E) \to Per_i$ *is an observation function for* $\mathsf{Ag}_i$, *mapping the state of all agents and the environment to a percept of the agent, implemented via an NN classifier;*

- $\delta_i : S_i \times A \to \mathbb{P}(Loc_i)$ *is a probabilistic transition function for* $\mathsf{Ag}_i$, *where* $\mathbb{P}(X)$ *denotes the set of probability distributions over a set* $X$, *determining the probability of moving to local states given its current state and joint action;*

- $\delta_E : S_E \times A \to S_E$ *is a deterministic environment transition function determining the environment's next state given its current state and joint action.*

Each (global) state $s$ of NS-CSG C comprises the state $s_i = (loc_i, per_i) \in S_i$ of each agent $\mathsf{Ag}_i$ and the state $s_E \in S_E$ of the environment. Starting from some initial state, the game evolves as follows. First, each agent $\mathsf{Ag}_i$ observes the state of the agents and the environment to generate a new percept $per'_i$ according to its observation function $obs_i$ implemented via an NN. Then, each agent $\mathsf{Ag}_i$ synchronously chooses one of the actions from the set $\Delta_i(s_i)$, which are available in its state $s_i$. This results in a joint action $\alpha = (a_1, \ldots, a_n) \in A$. Each agent $\mathsf{Ag}_i$ then updates its local state to $loc'_i \in Loc_i$ according to the probabilistic local transition function $\delta_i$, applied to the state of agent $(loc_i, per'_i)$ and joint action $\alpha$. The environment updates the environment state to $s'_E \in S_E$ according to the environment transition function $\delta_E$, applied to its state $s_E$ and joint action $\alpha$. Thus, the game reaches the state $s' = (s'_1, \ldots, s'_n, s'_E)$, where $s'_i = (loc'_i, per'_i)$ for $i \in N$. For simplicity, we consider here deterministic environments, but the results can be directly extended to discrete probabilistic environments with finite branching.

For brevity, we omit the formal semantics of an NS-CSG, which can be found in [Yan et al., 2022b]. In fact, in this paper we consider a slight variant, differing in the point at which observations are made during each transition.

NS-CSGs are a subclass of continuous-state CSGs, which assume a particular structure for the transition function, distinguishing between agent and environment states and using an NN-based observation function to characterise which environment states have the same characteristics. This provides a trade-off between exploiting the full generality

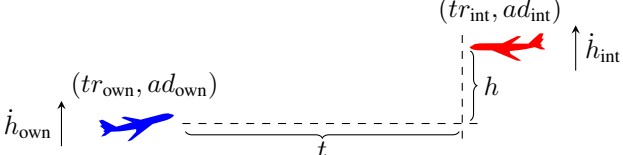

Figure 1: Geometry for the VCAS[2] example with trust level $tr_i$ and advisory $ad_i$, for $i \in \{\text{own}, \text{int}\}$.

of a continuous-state CSG model and allowing for tractable computational methods for its analysis.

Our use of NNs as perception functions to yield observations is in line with a recent trend in autonomous systems, where agents make decisions based on the output of NNs, for instance, probabilistic observation functions extracted from NNs by abstracting them with the help of robustness verification tools [Calinescu et al., 2022].

To illustrate NS-CSGs, we model the VerticalCAS Collision Avoidance Scenario [Julian and Kochenderfer, 2019, Julian et al., 2019] presented as a two-agent neurosymbolic system (VCAS[2]) in [Akintunde et al., 2020a]. Our model differs in that we separate the states of the agents and the environment state by adding to the agents' states a variable that measures their trust in the advisory's output, whereas [Akintunde et al., 2020a] replicates the climb rates in both agents' local states and the environment state. We update the agents' trust level probabilistically to account for possible uncertainty.

**Example 1.** *In the VCAS[2] system (Figure 1) there are two aircraft (ownship and intruder:* $\mathsf{Ag}_i$ *for* $i \in \{\text{own}, \text{int}\}$), *each of which is equipped with an NN-controlled collision avoidance system called VCAS. Each second, VCAS issues an advisory* $(ad_i)$ *from which, together with the current trust level* $(tr_i)$ *in the previous advisory, the pilot needs to make a decision about accelerations, aiming at avoiding a near mid-air collision (NMAC) [Akintunde et al., 2020b].*

*The input of the VCAS is* $(h, \dot{h}_{\text{own}}, \dot{h}_{\text{int}}, t)$ *recording the relative altitude* $h$ *of two aircraft, the climb rate* $\dot{h}_{\text{own}}$ *of the ownship, the climb rate* $\dot{h}_{\text{int}}$ *of the intruder, and the time* $t$ *until loss of their horizontal separation. VCAS is implemented via nine feed-forward NNs* $F = \{f_i : \mathbb{R}^4 \to \mathbb{R}^9 \mid i \in [9]\}$, *each of which corresponds to an advisory and outputs the scores of nine possible advisories, where* $[k]$ *is the set* $\{1, \ldots, k\}$. *Each advisory will provide a set of accelerations for the agent to select from. There are four trust levels* $\{4, 3, 2, 1\}$ *indicating the trust scores. The trust level is increased probabilistically if the current advisory is compliant with the executed action, and decreased otherwise. We formulate VCAS[2] as an NS-CSG with the agents* $\mathsf{Ag}_i$ *for* $i \in \{\text{own}, \text{int}\}$ *and the environment defined as follows:*

- $s_i = (tr_i, ad_i)$ *is a state of the agent* $\mathsf{Ag}_i$ *with local state* $tr_i \in [4]$ *and percept* $ad_i \in [9]$;

- $s_E = (h, \dot{h}_{\text{own}}, \dot{h}_{\text{int}}, t)$ *is an environment state;*

- $A_i$ is a finite set of accelerations ($\ddot{h}_i$);

- $\Delta_i(s_i)$ returns a set of available accelerations;

- observation function $obs_i$ is implemented via $F$;

- the local transition function $\delta_i$ updates its trust level according to its current trust level, its updated advisory and its executed action;

- the environment transition function $\delta_E(s_E, \alpha)$ is defined as: $h' = h - \Delta t(\dot{h}_{\text{own}} - \dot{h}_{\text{int}}) - 0.5\Delta t^2(\ddot{h}_{\text{own}} - \ddot{h}_{\text{int}})$, $\dot{h}'_{\text{own}} = \dot{h}_{\text{own}} + \ddot{h}_{\text{own}}\Delta t$, $\dot{h}'_{\text{int}} = \dot{h}_{\text{int}} + \ddot{h}_{\text{int}}\Delta t$ and $t' = t - \Delta t$, where $\Delta t = 1$ is the time step.

**Game Tree Unfolding.** The finite-horizon evolution of an NS-CSG C from a given global state $s$ can be unfolded into a finite tree in the usual way by applying *strategies* to select actions. We distinguish between (past) *histories* of a given state and its (future) *paths*.

We assume that the duration of the game is finite with $K$ stages. A history $h$ of C in stage $\ell \in [0, K]$ is a sequence $h = s^0 \xrightarrow{\alpha^0} s^1 \xrightarrow{\alpha^1} \cdots \xrightarrow{\alpha^{\ell-1}} s^\ell$ where $s^k \in S$, $\alpha^k \in A$ and $\delta(s^k, \alpha^k)(s^{k+1}) > 0$. The prefix of $h$ ending in stage $\bar{\ell}$ is denoted by $h_{\leq \bar{\ell}}$ for any $\bar{\ell} \leq \ell$. The set of all histories in stage $\ell$ for all initial states (for an initial state $s$) is denoted by $H^\ell$ ($H_s^\ell$), the set of all histories before stage $K$ is $H^{<K} = \cup_{0 \leq \ell < K} H^\ell$ ($H_s^{<K} = \cup_{0 \leq \ell < K} H_s^\ell$) and the set of all histories from $s$ is $H_s = H_s^{<K} \cup H_s^K$. We denote by $last(h)$ the last state of the history $h \in H_s$. If $h \in H^{<K}$, we denote by $\text{Succ}(h)$ the set of one-stage successors of $h$.

For a state $s = (s_1, \ldots, s_n, s_E)$, the available actions of $\text{Ag}_i$ are denoted by $A_i(s)$, i.e., $A_i(s)$ equals $\Delta_i(s_i)$ if $\Delta_i(s_i) \neq \varnothing$ and equals $\{\bot\}$ otherwise, and we denote by $A(s)$ the possible joint actions in a state, i.e. $A(s) = A_1(s) \times \cdots A_n(s)$.

We can now define strategies, strategy profiles and correlated profiles. In each case, we follow [Yan et al., 2022b] in assuming a *fully observable* setting as a baseline, i.e., where decisions are made based on the full state of the NS-CSG, not just the parts of it revealed by the agents' observation functions. An extension to partial observability (i.e., where the NS-CSG represents a continuous-state partially observable stochastic game) is left for future work.

**Definition 2** (Strategy). *A strategy for $\text{Ag}_i$ is a function $\sigma_i : H^{<K} \to \mathbb{P}(A_i \cup \{\bot\})$ such that, if $\sigma_i(h)(a_i) > 0$, then $a_i \in A_i(last(h))$. A strategy profile $\sigma = (\sigma_1, \ldots, \sigma_n)$ comprises a strategy for each agent. We denote by $\Sigma_i^{\text{N}}$ the set of all strategies for $\text{Ag}_i$ and by $\Sigma^{\text{N}} = \Sigma_1^{\text{N}} \times \cdots \times \Sigma_n^{\text{N}}$ the set of all strategy profiles.*

Alternatively, we can use a *correlated profile*, in which agent choices are correlated. For brevity, we refrain from formally defining a correlation mechanism (such as public signals) and map directly to joint actions.

**Definition 3** (Correlated profile). *A correlated profile is a function $\tau : H^{<K} \to \mathbb{P}(A)$ such that if $\tau(h)(\alpha) > 0$, then $\alpha = (a_1, \ldots, a_n)$ and $a_i \in A_i(last(h))$ for all $i \in N$. We denote by $\Sigma^{\text{C}}$ the set of correlated profiles.*

A (future) path $\pi$ of C starting from a history $h \in H^\ell$ in stage $\ell$ until the game ends in stage $K$ is a sequence $\pi = s^\ell \xrightarrow{\alpha^\ell} \cdots \xrightarrow{\alpha^{K-1}} s^K$ where $s^\ell = last(h)$, $s^k \in S$, $\alpha^k \in A$ and $\delta(s^k, \alpha^k)(s^{k+1}) > 0$. For path $\pi$, $\pi(k)$ is the $(k+1)$th state, $\pi[k]$ the action associated with the $(k+1)$th transition from $\pi(k)$ to $\pi(k+1)$, and $last(\pi)$ the final state.

**Rewards.** We endow NS-CSGs with *rewards* that define agents' objectives. We use $r = (r_i)_{i \in N}$ where each agent $\text{Ag}_i$ has a reward structure $r_i = (r_i^A, r_i^S)$ comprising action reward function $r_i^A : S \times A \to \mathbb{R}$ and state reward function $r_i^S : S \to \mathbb{R}$. An *objective profile* is $Y = (Y_1, \ldots, Y_n)$, where $Y_i(\pi)$ is the accumulated reward of $\text{Ag}_i$ until the final stage $K$, along a path $\pi$ that starts in some stage $\ell \in [0, K]$:

$$Y_i(\pi) = \sum_{k=0}^{K-\ell-1} \left( r_i^A(\pi(k), \pi[k]) + r_i^S(\pi(k)) \right) + r_i^S(last(\pi)).$$

Given a strategy profile $\sigma \in \Sigma^{\text{N}}$, we denote by $\mathbb{E}_{\ell,h}^\sigma[Y_i]$ the expected value of $Y_i$ when starting from $h \in H^\ell$ at the $\ell$th stage until the game ends. Given a correlated profile $\tau \in \Sigma^{\text{C}}$, we denote by $\mathbb{E}_\ell^\tau[Y_i, a_i'|a_i, h]$ the expected value of $Y_i$ when starting from $h \in H^\ell$ at the $\ell$th stage until the game ends, under the strategy that $\text{Ag}_i$ takes the actual action $a_i'$ instead of the recommended action $a_i$ at $h$, and otherwise the recommendation by $\tau$ is followed by all agents.

An NS-CSG is *zero-sum* if $\sum_{i=1}^n \left( r_i^A(s, \alpha) + r_i^S(s) \right) = 0$ for all $s \in S$ and all $\alpha \in A$; otherwise, it is *nonzero-sum*.

**Social Welfare Subgame-Perfect Equilibria.** A *Nash equilibrium* (NE) ensures that no agent has an incentive to deviate unilaterally from their strategy. Here we work with *subgame-perfect Nash equilibria* (SPNEs) [Osborne et al., 2004], which are NEs in every state of the game. Since an SPNE is therefore an NE of every subgame of the original game, the agents' behaviour from any point in the game onward forms an NE of the continuation game, regardless of what happened before. We also consider the less well studied notion of *subgame-perfect correlated equilibria* (SPCEs) [Murray and Gordon, 2007]. For an SPCE, no agent can expect to gain by disobeying the recommendation of the correlated profile after any history of play.

The formal definitions of both types of subgame-perfect equilibria (SPE) follow, where we denote by $\mu = \mu_{-i}[\mu_i] = (\mu_1, \ldots, \mu_n)$ ($i \in N$) the strategy profile, where $\mu_{-i}$ refers to the strategy profile except $\mu_i$. For SPCEs, we again omit a correlation mechanism and abuse notation by expressing it as individual deviations from the recommended actions associated to a correlated profile $\tau$.

**Definition 4** (Subgame-perfect equilibrium). *For an initial state $s \in S$, a strategy profile $\sigma^* = (\sigma_1^*, \ldots, \sigma_n^*) \in \Sigma^N$ is a subgame-perfect Nash equilibrium (SPNE) if $\mathbb{E}_{\ell,h}^{\sigma^*}[Y_i] \geq \mathbb{E}_{\ell,h}^{\sigma_{-i}^*[\sigma_i]}[Y_i]$ for all $\sigma_i \in \Sigma_i^N$, all $i \in N$ and all $h \in H_s^{<K}$. A correlated profile $\tau^* \in \Sigma^C$ is a subgame-perfect correlated equilibrium (SPCE) if $\mathbb{E}_{\ell}^{\tau^*}[Y_i, a_i | a_i, h] \geq \mathbb{E}_{\ell}^{\tau^*}[Y_i, a_i' | a_i, h]$ for all $a_i, a_i' \in A_i(last(h))$, all $i \in N$ and all $h \in H_s^{<K}$.*

We emphasize that the SPE is defined here for a given initial state. Since multiple SPEs can exist, we introduce additional optimality constraints. First, we define the *social welfare* $W_{\ell,h}^{\sigma}$ ($W_{\ell,h}^{\tau}$, resp.) of a history $h \in H^{\ell}$ ($\ell < K$) under a strategy profile $\sigma$ (a correlated profile $\tau$, resp.) as the sum of expected values of objective profiles $Y_i$ starting in $h$ for all agents, that is, $W_{\ell,h}^{\sigma} = \mathbb{E}_{\ell,h}^{\sigma}[\sum_{i=1}^{n} Y_i]$ ($W_{\ell,h}^{\tau} = \mathbb{E}_{\ell,h}^{\tau}[\sum_{i=1}^{n} Y_i]$, resp.). Social-welfare optimal SPNE and and SPCE are then defined as follows.

**Definition 5** (Social welfare SPE). *For an initial state $s \in S$, an SPNE $\sigma^*$ is a social welfare optimal SPNE (SW-SPNE) of C if $W_{0,s}^{\sigma^*} \geq W_{0,s}^{\sigma}$ for all SPNEs $\sigma$ of C. An SPCE $\tau^*$ is a social welfare optimal SPCE (SW-SPCE) of C if $W_{0,s}^{\tau^*} \geq W_{0,s}^{\tau}$ for all SPCEs $\tau$ of C.*

Notice that, starting from a fixed initial state, SW-SPNE and SW-SPCE are *globally optimal*, i.e. over the social welfare achieved over a finite horizon from that start state.

Our approach of defining optimality in terms of the value from a fixed initial state is further motivated by the following result, which reveals that SW-SPNEs and SW-SPCEs do not possess the property of subgame perfection on social welfare, i.e., an SPNE or SPCE with optimal social welfare at one state might induce a non-optimal social welfare at another state as the game moves forward.

**Lemma 6** (No optimal subgame perfection). *For an initial state $s \in S$, an NS-CSG may have no SPNE (resp., SPCE) that is an SW-SPNE (resp., SW-SPCE) for all its subgames.*

A proof of this, and all other results in the paper can be found in the appendix. Note also that this and the following results are stated in the context of NS-CSGs, but they also apply to general CSGs with discrete states and actions.

## 3 GENERALIZED BI

We now consider how to *compute* equilibria for NS-CSGs. For a fixed initial state, finite-horizon NS-CSGs are finite games, obtained by unfolding the game tree while invoking the NN perception function. In principle, this allows us to employ established game-theoretic solution such as backward induction. We next prove that the classical *generalized backward induction* (GBI) [Shoham and Leyton-Brown, 2009] can be used to find a finite-horizon SPNE or SPCE

---

**Algorithm 1** Generalized b/w induction (GBI) via SWE

**Input:** NS-CSG C, rewards $r$, equ. type T, initial state $s$
**Output:** an equilibrium $\mu$, equilibrium payoff vector $V$

1: $H_s^{\ell} \leftarrow \text{HISTORY}(C, s, \ell)$ for all $\ell \leq K$
2: **for** $\ell = K, K-1, \ldots, 0; h \in H_s^{\ell}$ **do**
3:    **if** $\ell = K$ **then**
4:       $V^h \leftarrow (r_1^S(last(h)), \ldots, r_n^S(last(h)))$
5:    **else**
6:       $\text{Succ}(h) \leftarrow \text{SUCCESSOR}(C, H_s^{\ell+1}, h)$
7:       $(\mu^h, V^h) \leftarrow \text{SWE\_SOLVER}(C, r, T, , h,$
                               $\{V^{h'} \mid h' \in \text{Succ}(h)\})$
8: $\mu \leftarrow \{\mu^h\}_{h \in H_s^{<K}}, V \leftarrow \{V^h\}_{h \in H_s}$
9: **return** $\mu, V$

---

through local optimisation, but that this equilibrium might have an arbitrarily bad social welfare.

Algorithm 1 shows a version of the classical GBI method, for concurrent extensive-form games over a finite horizon, which aims to find an SPNE or SPCE that maximises social welfare, by computing an NE or CE which is *locally* social welfare maximal at each history. In Algorithm 1, $\text{HISTORY}(C, s, \ell)$ computes a set of all histories in stage $\ell$ given an initial state $s \in S$. $\text{SUCCESSOR}(C, H_s^{\ell+1}, h)$ extracts a set of all successors of a history $h$ in stage $\ell$ from $H_s^{\ell+1}$. $\text{SWE\_SOLVER}(C, r, T, , h, \{V^{h'} \mid h' \in \text{Succ}(h)\})$ computes an SWNE or SWCE $\mu^h$ (depending on the equilibrium type $T \in \{CE, NE\}$) of an induced normal-form game with actions available at $last(h)$ and utilities from the equilibrium payoffs $V^{h'}$ of all successors $h'$ of $h$, and then assigns the equilibrium payoff associated with this equilibrium to $V^h$. This procedure is iterated from the bottom up until $\ell = 0$, i.e., $h = s$, where the equilibrium payoffs of histories at stage $K$ (i.e., where the game ends) are equal to final states' rewards. For this algorithm, we have the following proposition.

**Proposition 7.** *Given an initial state $s \in S$, GBI finds an SPNE $\sigma$ (SPCE $\tau$, resp.) with social welfare $W_{0,s}^{\sigma} = \sum_{i \in N} V_i^s$ ($W_{0,s}^{\tau} = \sum_{i \in N} V_i^s$, resp.).*

Although GBI can find an SPNE or SPCE, unfortunately it may return one with an arbitrarily bad social welfare with respect to the optimum.

**Lemma 8** (Bad social welfare). *The SPNE (SPCE, resp.) obtained by GBI SWE can be arbitrarily bad on social welfare with respect to an SW-SPNE $\sigma^*$ (SW-SPCE $\tau^*$, resp.) for some state $s \in S$, i.e., $W_{0,s}^{\sigma^*} - W_{0,s}^{\sigma}$ ($W_{0,s}^{\tau^*} - W_{0,s}^{\tau}$, resp.) is positive and unbounded.*

## 4 FROZEN SUBGAME IMPROVEMENT

Lemma 8 indicates that a GBI-based approach does not guarantee optimal social welfare. Motivated by this, we now

consider further techniques to synthesize SW-SPNE and SW-SPCE for NS-CSGs. We first present an *exact* approach based on an unfolding of the game tree and the solution of a *nonlinear program*. However, this does not scale to large games. So we then propose an iterative *approximation* method called *frozen subgame improvement*. This works by first finding an arbitrary initial SPNE or SPCE and then iteratively freezing a set of variables and computing a new SPNE or SPCE with an increasing social welfare.

In this section, we focus initially on the case of *two-agent* NS-CSGs and then later discuss how to generalise this.

**Exact Computation of SW-SPNE and SW-SPCE.** Given an initial state $s \in S$, the game unfolds by considering all paths, thus generating a game tree which can be fully characterized by $H_s$. During the game tree construction, $last(h)$ can be computed for any $h \in H_s$, and if $h'$ is a successor of $h$, the joint action(s) that leads to $h'$ from $h$ can be determined. In contrast to [Akintunde et al., 2020a], where perception functions are assumed to be piecewise linear and encoded as constraints, unfolding the game tree allows us to treat NNs outside the optimisation problem.

We encode subgame perfection as a nonlinear program. An SPNE of the original game is an NE of every subgame, i.e., for each history $h \in H_s^{<K}$, it can be encoded as follows [1]:

$$V_i^h - \sum_{(a_i,a_j) \in A_i \times A_j} \mu_i^h(a_i) \cdot \mu_j^h(a_j) \cdot Z_i^{h,(a_i,a_j)} = 0$$
$$V_i^h - \sum_{a_j \in A_j} \mu_j^h(a_j) \cdot Z_i^{h,(a_i,a_j)} \geq 0, \ \forall a_i \in A_i \quad (1)$$
$$\sum_{a_i \in A_i} \mu_i^h(a_i) = 1, \ \mu_i^h(a_i) \geq 0$$

for $i,j \in \{1,2\}, i \neq j$, where $\mu_i^h \in \mathbb{P}(A_i(last(h)))$, $V^h = (V_1^h, V_2^h) \in \mathbb{R}^2$ denotes the expected accumulated reward vector from $h$ to the end of the game, and $Z_i^{h,\alpha}$ denotes the expected accumulated reward to be received by $Ag_i$ after executing the joint action $\alpha$ at $h$. In an SPCE, no agent can gain by deviating from the recommendation in any given history, and thus we have:

$$V_i^h - \sum_{\alpha \in A} \mu_\alpha^h \cdot Z_i^{h,\alpha} = 0$$
$$\sum_{a_j \in A_j} (Z_i^{h,(a_i,a_j)} - Z_i^{h,(a_i',a_j)}) \cdot \mu_{(a_i,a_j)}^h \geq 0 \quad (2)$$
$$\sum_{\alpha \in A} \mu_\alpha^h = 1, \quad \mu_\alpha^h \geq 0$$

where $i,j \in \{1,2\}, i \neq j, a_i, a_i' \in A_i$, $\mu^h = \{\mu_\alpha^h\}_{\alpha \in A}$ and $\mu_\alpha^h$ represents the probability of the joint action $\alpha$ being recommended at $h$.

The SPNE and SPCE imply that, for each $h \in H_s^{<K}$ and $\alpha \in A(last(h))$, the reward for $Ag_i$ satisfies:

$$Z_i^{h,\alpha} = r_i^A(last(h), \alpha) + r_i^S(last(h))$$
$$+ \sum_{h' \in Succ(h)} \delta(last(h), \alpha)(last(h')) V_i^{h'} \quad (3)$$

---

[1] To simplify notation, $a_i \in A_i$ refers to $a_i \in A_i(last(h))$ in (1) and (2), and similarly for $a_j$ and $a_i'$.

where, for each history $h \in H_s^K$, we take the reward vector $V^h = (r_1^S(last(h)), r_2^S(last(h)))$. For each $h \in H_s^{<K}$, let $C^{N,h}(\mu_1^h, \mu_2^h, V^h, \{V^{h'}\}_{h' \in Succ(h)})$ be the union of constraints (1) and (3) (for Nash equilibria), and $C^{C,h}(\mu^h, V^h, \{V^{h'}\}_{h' \in Succ(h)})$ be the union of constraints (2) and (3) (for correlated). The union of $C^{N,h}$ for all such histories is denoted by $C^N(\mu^N, V)$ and the union of $C^{C,h}$ by $C^C(\mu^C, V)$, where $\mu^N := \{\mu_1^h, \mu_2^h\}_{h \in H_s^{<K}}$, $\mu^C := \{\mu^h\}_{h \in H_s^{<K}}$ and $V := \{V^h\}_{h \in H_s^{<K}}$. Note that $C^N(\mu^N, V)$ ($C^C(\mu^C, V)$, resp.) is polynomial in $\mu^N$ ($\mu^C$, resp.) and $V$, and is nonlinear as $Z_i^{h,\alpha}$ is related to variables $V_i^{h'}$ for $h' \in Succ(h)$.

**Theorem 9** (Computation of SW-SPNE and SW-SPCE). *For a two-agent NS-CSG C with an initial state $s \in S$,*

*(i) a strategy profile $\sigma$ is an SPNE iff there is a solution of the constraints $C^N(\mu^N, V)$ such that $\sigma_1(h) = \mu_1^h$ and $\sigma_2(h) = \mu_2^h$ for each $h \in H_s^{<K}$;*

*(ii) a correlated profile $\tau$ is an SPCE iff there is a solution of the constraints $C^C(\mu^C, V)$ such that $\tau(h) = \mu^h$ for each $h \in H_s^{<K}$;*

*(iii) a strategy profile $\sigma$ is an SW-SPNE iff there is an optimal solution $(\mu^*, V^*)$ of the nonlinear program:*

$$\max_{\mu^N, V} \ \sum_{i \in N} V_i^s \qquad \text{subject to} \quad C^N(\mu^N, V) \quad (4)$$

*such that $\sigma_1(h) = \mu_1^{*,h}$ and $\sigma_2(h) = \mu_2^{*,h}$ for each $h \in H_s^{<K}$, and the social welfare $W_{0,s}^\sigma$ is equal to the optimal value $\sum_{i \in N} V_i^{*,s}$;*

*(iv) a correlated profile $\tau$ is an SW-SPCE iff there is an optimal solution $(\mu^*, V^*)$ of the nonlinear program:*

$$\max_{\mu^C, V} \ \sum_{i \in N} V_i^s \qquad \text{subject to} \quad C^C(\mu^C, V) \quad (5)$$

*such that $\tau(h) = \mu^{*,h}$ for each $h \in H_s^{<K}$, and the social welfare $W_{0,s}^\tau$ is equal to the optimal value $\sum_{i \in N} V_i^{*,s}$.*

Although our goal here is to work with NNs, the computation of SW-SPNE and SW-SPCE in Theorem 9 also applies to conventional stochastic games, because the game tree construction can work for general transition functions with finite branching. The fact that our approach is not limited to NNs (or NNs of a certain class) is an advantage, and allows us to avoid the scalability issues suffered by the method of [Akintunde et al., 2020a], which represents a ReLU neural network as a set of constraints.

**Frozen Subgame Improvement.** Nonlinear programs in Theorem 9 can be used to find an SW-SPNE or SW-SPCE efficiently for a small joint action profile and a short horizon. For larger problems, scalability is an issue because the numbers of variables and constraints are both exponential.

To deal with this, we propose an approximation algorithm called *Frozen Subgame Improvement (FSI)* (Algorithm 2) that trades optimality for scalability.

---

**Algorithm 2** Frozen Subgame Improvement (FSI)

---

**Input:** NS-CSG C, reward $r$, equ. type T, init. state $s$, $m_{\max}$
**Output:** an equilibrium $\mu$, equilibrium payoff vector $V$

1: $(\mu, V) \leftarrow$ GENERALIZED_BI$(C, r, T, , s)$
2: $m \leftarrow 0$
3: **repeat**
4:      $h \leftarrow$ A_HISTORY$(H_s^{<K}, \mu, V)$
5:      $P \leftarrow$ (4) or (5) (depending on T) after freezing $\mu^{h'}, V^{h'}$ for each history $h' \in H_s^{<K}$ that is not a prefix of $h$ (say $h \in H_s^\ell$ for some $\ell < K$);
6:      $\{\mu^{*,h\le\bar{\ell}}, V^{*,h\le\bar{\ell}}\}_{\bar{\ell}\le\ell} \leftarrow$ NP_SOLVER$(P)$
7:      $\mu \leftarrow \{\mu^{*,h\le\bar{\ell}}\}_{\bar{\ell}\le\ell} \cup \{\text{the frozen } \mu^{h'}\}$
8:      $V \leftarrow \{V^{*,h\le\bar{\ell}}\}_{\bar{\ell}\le\ell} \cup \{\text{the frozen } V^{h'}\}$
9:      $m \leftarrow m + 1$
10: **until** $m = m_{\max}$
11: **return** $\mu, V$

---

The main idea of FSI is as follows. First, GBI is used to find a feasible solution to (4) or (5) depending on the equilibrium type $T \in \{CE, NE\}$, i.e., an SPNE or SPCE. Then, a history $h \in H_s^{<K}$ is selected, for example by sampling uniformly. We freeze the distributions over (joint) actions and equilibrium payoffs corresponding to the histories that are not prefixes of $h$. Thus, (4), and similarly (5), can be simplified into a nonlinear program with a smaller number of variables and constraints. Finally, a new solution is computed by merging the frozen part of the current solution and an optimal solution of the simpler nonlinear program. The process performs a predefined number $m_{\max}$ of iterations.

In Algorithm 2, GENERALIZED_BI$(\cdot)$ computes an SPNE or SPCE $\mu$ and the associated equilibrium payoff vector $V$ by adopting a simpler version of Algorithm 1, in which an NE or CE is computed at step 7 instead of an SWNE or SWCE. A_HISTORY$(\cdot)$ returns a history. Here, we sample a history from $H_s^{K-1}$ uniformly; an alternative is presented in Appendix. NP_SOLVER$(\cdot)$ computes an optimal solution to a given nonlinear program.

For FSI, we have the following results:

**Theorem 10** (FSI). *If FSI is adopted to solve* (4) *((5), resp.) approximately, then:*

(i) *the pair $(\mu, V)$ is a feasible solution to* (4) *((5), resp.) at the end of each iteration $m$, that is, $\mu$ is an SPNE (SPCE, resp.) and $V$ is the equilibrium payoff vector;*

(ii) *the social welfare $\sum_{i\in N} V_i^s$ is monotonically increasing in $m$, and also monotonically increasing in $m_{\max}$.*

**FSI over Regions.** If each agent has a limited memory and takes actions conditioned on the current state and stage, we

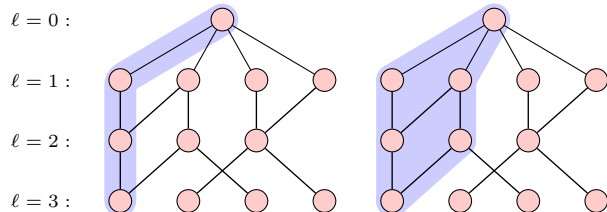

Figure 2: FSI over regions. Sampled history (left) and the corresponding region (right).

can unfold the game into a graph where each node in a stage represents one reachable state exactly in that stage, as in Fig. 2. With respect to the game tree, the number of nodes in this graph is greatly decreased if many states are frequently visited in a stage. The FSI can be directly adapted to this graph by first sampling a history (Fig. 2: left) and then optimising over a region of states, which contain all histories that reach its last state (Fig. 2: right).

**Multi-agent.** SW-SPNE and SW-SPCE computation for *multi-agent* ($n>2$) NS-CSGs can be performed by replacing (1) or (2) with the encoding of NE/CE computation for the induced multi-agent normal-form game at each $h \in H_s^{<K}$.

**Complexity.** We focus here on practical methods to compute equilibria, which depend on the horizon $K$ and the size of the model (specifically the number of actions and agent states), as well as the underlying solution method used to solve either normal form games (at each state, for SWNE or SWCE) or nonlinear optimisation problems (for SW-SPNE or SW-SPCE). Computing NEs of a normal form game with two players is known to be PPAD-complete [Chen et al., 2009]. For extensive games, it has been proved that finding SPNEs for quantitative reachability objectives of a two-player game is PSPACE-complete [Brihaye et al., 2019]. Computing SWCEs of a normal form game can be done in polynomial time [Gilboa and Zemel, 1989].

From a practical perspective, any method that relies on finding all NEs in the worst case cannot be expected to achieve a running time that is polynomial with respect to the size of the game, as there can be exponentially many equilibria. GBI requires us to compute an SWNE or SWCE for all states that could be reached from a given initial state in $K$ steps. FSI relies on GBI as an initialisation step (Algorithm 2, line 1). Furthermore, the optimisation problem defined for computing SW-SPNE in (4) has at most $(|A_1| + |A_2| + 2)v$ variables and $(2|A_1||A_2| + 2|A_1| + 2|A_2| + 4)v$ constraints, and for computing SW-SPCE defined in (5) has at most $(|A_1||A_2| + 2)v$ variables and $(|A_1||A_2| + |A_1|^2 + |A_2|^2 - |A_1| - |A_2| + 3)v$ constraints, where $v$ is the number of non-leaf nodes in the generated game tree and $v = ((|A_1||A_2||S_1||S_2|)^K - 1)/(|A_1||A_2||S_1||S_2| - 1)$ in the worst case.

# 5 EXPERIMENTAL EVALUATION

We have implemented a prototype version of our FSI method (Algorithm 2). This uses components from PRISM-games 3.0 [Kwiatkowska et al., 2020], which supports discrete CSGs without perception. In particular, we use its SMT-based/linear programming method for synthesising CSG SWNE/SWCE to initialise the vector of equilibria values in line 1 of Algorithm 2. Its support for two-player finite-horizon equilibria [Kwiatkowska et al., 2019] also gives an equivalent version of the GBI algorithm (Algorithm 1).

The optimisation problems for computing SW-SPNE and SW-SPCE values for states are solved using Gurobi. In order to improve the scalability of FSI, our implementation considers a reduced set of histories by: (i) limiting the information that the players have access to at each state to be the values of the variables in that state plus *time*, i.e., how many transitions have been made up until that point; and (ii) constructing histories not over states, but *regions* of states which are independent from a decision-making standpoint.

Our evaluation employs two case studies: the first is used to show the applicability of our equilibria improvement algorithm, and the second to demonstrate the usefulness of equilibria properties for analysing NS-CSGs. An overview is provided below, with more detail given in the appendix.

**Automated Parking.** We first formulate a dynamic vehicle parking problem as an NS-CSG (a static assignment game is considered in, e.g., [Ayala et al., 2011]). There are 2 players (vehicles) targeting 2 parking slots in a $5 \times 4$ grid, shown in Fig. 3 (target cells are green, forbidden cells are red, black arrows show traffic rules). We consider two reward structures. One minimises time, while the other extends the first by giving a bonus to player 2 for visiting a designated cell (in yellow). This is a discrete-state model in which percepts identify agent locations precisely. We use it to compare the equilibria algorithms for two different time horizons $K = 8$ and $K = 6$. For this model, both vehicles get a reward of -1 for each move, vehicle 2 gets a reward of 5.5 when visiting the bonus cell and the speeds of vehicle 1 and 2 are of two and one grid cell per move, respectively.

We first consider Nash equilibria. For the first reward structure, our FSI algorithm and the GBI algorithm, which only considers local SWNE values, both return the SW-SPNE strategy with reward sum $-5.0$ in Fig. 3 (top-left). For the second reward structure, FSI finds a new SW-SPNE strategy with reward sum $-4.5$ in Fig. 3 (top-right) giving a higher social welfare, while GBI still returns the strategy on the left, which is not an SW-SPNE in this case.

With correlated equilibria, for $K = 8$ both algorithms produce the same strategy as in Fig. 3 (bottom-right), for which the reward sum is -1.5. We then reduce the time horizon to $K = 6$. For this case, in the strategy constructed by the

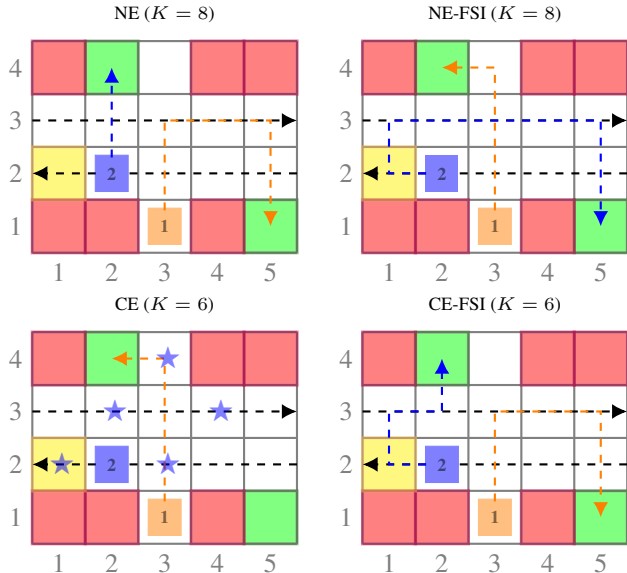

Figure 3: Strategies for the automated parking example.

| $K$ | States | Trans. | Constr. time (s) | GBI time (s) | | Region size | | FSI time (s) | |
|---|---|---|---|---|---|---|---|---|---|
| | | | | NE | CE | NE | CE | NE | CE |
| 6 | 258 | 1080 | 0.01 | 0.6 | 2.1 | 24.0% | 22.5% | 0.4 | 1.5 |
| | | | | | | 19.4% | 20.2% | 0.4 | 1.0 |
| | | | | | | 17.8% | 16.3% | 0.2 | 0.3 |
| 8 | 386 | 1689 | 0.2 | 1.4 | 4.9 | 37.3% | 32.4% | 3.8 | 2.5 |
| | | | | | | 32.4% | 27.5% | 1.8 | 2.6 |
| | | | | | | 25.9% | 25.9% | 1.1 | 1.8 |

Table 1: Statistics for the automated parking example.

GBI algorithm in Fig. 3 (bottom-left), vehicle 2 is instructed to move left in order to get the bonus, while vehicle 1 is instructed to park in the closest spot. However, given the shorter horizon, vehicle 2 does not have enough time to park in the remaining spot and the overall reward sum is -2.5. The possible final positions for vehicle 2 are indicated by the blue stars. In the strategy synthesised by the FSI algorithm, however, both cars park and the sum of rewards is higher. Table 1 shows statistics for the models constructed and the time for equilibria computation.

**Two-Agent Aircraft Collision Avoidance Scenario.** Secondly, we consider an NS-CSG model of the VCAS[2] system, as described earlier in Example 1. We study its equilibria strategies, in contrast to the zero-sum (reachability) properties analysed in [Akintunde et al., 2020a]. Fig. 4 plots the altitude $h$ for equilibria and zero-sum strategies when maximising $h$ for a given instant $k$. It can be seen that, with respect to the safety criterion established by [Akintunde et al., 2020a, Julian and Kochenderfer, 2019], i.e., avoiding a near mid-air collision, equilibria strategies allow the two aircraft to reach a safe configuration within a shorter horizon, which would be missed by a zero-sum analysis.

We also consider a second reward structure that incorporates the trust level and fuel consumption, and we vary the agent

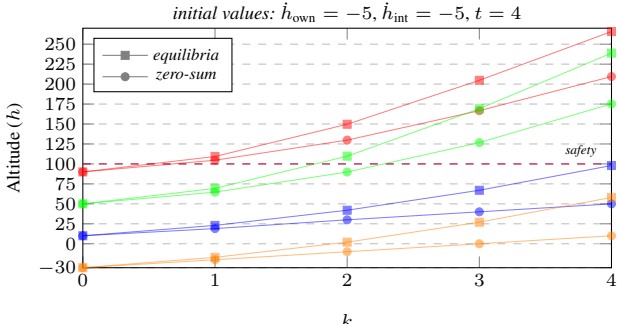

Figure 4: Altitude ($h$) for the VCAS[2] example.

| $\epsilon_{own}, \epsilon_{int}$ | $t$ | States | Constr. time (s) | GBI time (s) NE | GBI time (s) CE | $h$ | Viol. |
|---|---|---|---|---|---|---|---|
| 0, 0 | 2 | 100 | 0.06 | 0.1 | 0.05 | 82 | 0 |
| | 3 | 836 | 0.6 | 0.7 | 0.3 | 123 | 0 |
| | 4 | 6997 | 36.6 | 8.0 | 1.8 | 199 | 25% |
| 0.1, 0 | 2 | 157 | 0.1 | 0.2 | 0.1 | 82 | 0 |
| | 3 | 1622 | 1.4 | 1.0 | 0.3 | 123 | 0 |
| | 4 | 16028 | 273.8 | 14.2 | 3.3 | 199 | 20% |
| 0.1, 0.2 | 2 | 251 | 0.1 | 0.2 | 0.07 | 82 | 0 |
| | 3 | 3174 | 4.4 | 1.5 | 0.6 | 123 | 0 |
| | 4 | 36639 | 1497.2 | 26.7 | 5.8 | 199 | 20% |

Table 2: Statistics for the VCAS[2] example.

uncertainty parameters $\epsilon_i$ (see the appendix for details). We also fix a different safety limit of $h = 200$. Table 2 shows the altitude and number of violations (times that no advisory is taken) for the generated equilibria. To give an indication of scalability and performance, we also include the total number of states in the game unfolding and the time for model construction and algorithm execution for both NE and CE. For this example, both types of equilibria yield the same values for the properties considered.

Finally, we discuss equilibria strategies for different values of the uncertainty parameter $\epsilon_{own}$. We find that the agents always comply with the advisory system for smaller initial values of $t$ (time until loss of horizontal separation), given that reaching safety would be of higher priority. Fig. 5 (left) illustrates that following the advisories is the best strategy when safety and trust are the priority, as the trust levels $tr_{own}$ and $tr_{int}$ of the two agents never decrease from the initial score of $4$. This changes, however, when both aircraft have a larger horizon to consider. The strategy in Fig. 5 (right) shows a deviation from the advisory (denoted by value 0 for $a_{own}$ in state $s^2$), resulting in $tr_{own}$ dropping to 3 in $s^3$ with probability $0.9$, reduced fuel consumption and the safety limit of 200 being approached.

**Efficiency and scalability.** For equilibria computation using GBI, which computes locally optimal equilibria, CE are generally considerably faster to compute than NE. This is due to the fact that finding an optimal CE in a state can be reduced to solving a *linear program*, while computing an optimal NE requires finding all solutions of a *linear complementarity problem*. The same, however, is not observed when comparing the performance of FSI on the two types of equilibria. This is because a path-based encoding requires a greater number of constraints and variables for CE, and we need to solve nonlinear programs.

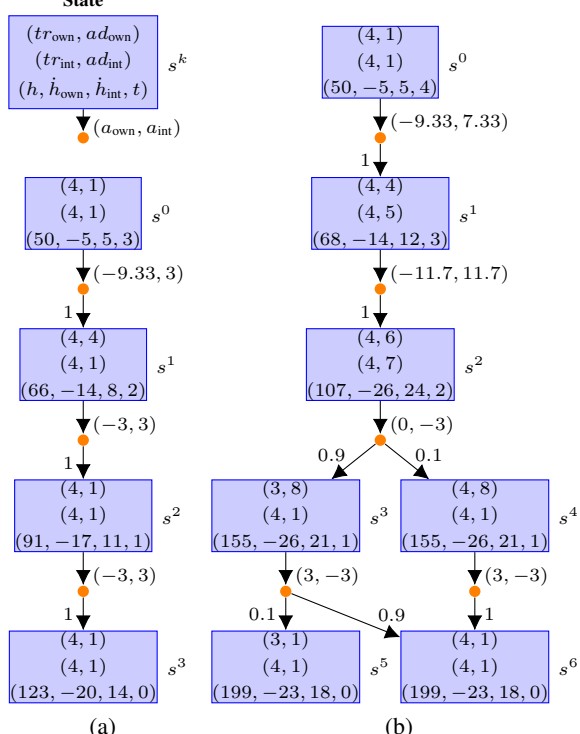

Figure 5: Strategies for the VCAS[2] example: (a) $\epsilon_{own} = 0$, $\epsilon_{int} = 0$ and $t$ initially 3; (b) $\epsilon_{own} = 0.1$, $\epsilon_{int} = 0$ and $t$ initially 4.

that improves on social welfare equilibria values and strategies, for both SPNE and SPCE, compared to backward induction, which can only reason about local optimality. A prototype implementation showcased its applicability and advantages on two case studies. Future work will focus on infinite-horizon properties (incorporating finite-horizon equilibria with receding horizon synthesis [Raman et al., 2015]) and temporal logic specifications.

# 6 CONCLUSIONS

We have considered finite-horizon equilibria computation for CSGs whose agents are equipped with NN-based perception mechanisms. We developed an approximate algorithm

**Acknowledgements**

This project was funded by the ERC under the European Union's Horizon 2020 research and innovation programme (FUN2MODEL, grant agreement No. 834115).

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
