# OpenReview forum: "Finite-horizon Equilibria for Neuro-symbolic Concurrent Stochastic Games"
_auai.org/UAI/2022/Conference — UAI 2022 Poster_

### Official Review · Reviewer_Z8Yu · 2022-04-03

**Q2(1) Originality/Novelty:** 2
**Q2(2) Significance/Impact:** 2
**Q2(3) Correctness/Technical Quality:** 3
**Q2(6) Clarity Of Writing:** 2
**Q6 Overall Score:** 6
**Q8 Confidence In Your Score:** 3

**Q1 Summary And Contributions:**

This paper presents the framework of neuro-symbolic comncurrent stochastic games and proposes some exact and approximate algorithms for computing Nash equilibria or correlated equilibria, maximizing a social welfare function for these games.
This paper also proposes experiments (without available code) and relies on a recent paper by Yan et al., 2022).

**Q10 Ethical Concerns (Optional):**

I do not know what to think about the possible breach of the anonimity rule (referring to an unpublished, very recently available paper using the same notations and experimental testbed). I think it is more an awkwardness than lack of honesty and would not see it as a reason for rejection, but this should be avoided.

**Q2 Assessment Of The Paper:**

More detailed information regarding each of these aspects is given below:

**Q2(4) Quality Of Experiments (Optional):**

2: Fair: The experimental evaluation is weak: important baselines are missing, or the results do not adequately support the main claims.

**Q2(5) Reproducibility:**

2: Fair: Key resources (e.g., proofs, code, data) are unavailable but key details (e.g., proof sketches, experimental setup) are sufficiently well-described for an expert to confidently reproduce the main results.

**Q3 Main Strengths:**

The problem of finding social welfare optimal equilibria in stochastic games is important.
The paper proposes some argumented mathematical results.
The methods have been experimented.

**Q4 Main Weakness:**

The exact original contribution of the paper with respect to the state of the art is hard to assess, due to (unnecessary?) complexity of exposure (see below) and of terms.
The paper's contribution heavily relies on a recent, unpublished paper (deposited on Feb 2022 on Arxiv) and the contribution is a bit incremental.
Furthermore, the notations are very close, one of the case study is the same, meaning that anonimity of the current contribution is questionable.


**Q5 Detailed Comments To The Authors:**

Even though I do not mean that the contribution of the paper is negligible, it is hard for me to indentify its impact.

First, the framework considered, that of NN-CSG raises questions (which I would like the authors to answer):
1) In what respect are CSG different from usual "Stochastic Games"? It looks like the difference could be partial observability, but then what is the difference with partially-observed stochastic games (POSG) framework, which is well-known too and has attracted a lot of interest?
2) What is the added value of considering Neural Networks for observation functions and entitling the method as neuro-symbolic? I may have missed something, but it looks like observations are just dealt with as "abstract" functions and the fact that they may be computed using NNs is irrelevant in the paper...

Besides the above two remarks, my feeling is that they authors consider very classical approaches (Backwards induction or what is called Frozen subgames improvement) to finding maximal (or approximate) social welfare equilibria in POSG.

Such approaches are not very different from existing approches in stochsatic games (but I may have missed sometihing).

**Q7 Justification For Your Score:**

The main reasons for my evaluation are that the writing of the paper is unnecessarily complex and does not help understanding. I don't really understand the originality of CSG with respoct to (PO)SG and I don't understand why the authors consider it important that a NN is used to model a classical finite observation probability table (which is used as such in the algorithms).
Once I have managed to peel off the unnecessary complexity, the results which remain seem to me very classical.

**Q9 Complying With Reviewing Instructions:**

1: Yes.

---

### Official Review · Reviewer_PLuj · 2022-04-12

**Q2(1) Originality/Novelty:** 3
**Q2(2) Significance/Impact:** 3
**Q2(3) Correctness/Technical Quality:** 3
**Q2(6) Clarity Of Writing:** 3
**Q6 Overall Score:** 8
**Q8 Confidence In Your Score:** 1

**Q1 Summary And Contributions:**

This paper introduces a novel technique for solving concurrent stocchastic games
in continuous-space environments. Various definitions of equilibria are used and
the paper seems relevant to the conference.




**Q2 Assessment Of The Paper:**

More detailed information regarding each of these aspects is given below:

**Q2(4) Quality Of Experiments (Optional):**

3: Good: The experimental evaluation is adequate, and the results convincingly support the main claims.

**Q2(5) Reproducibility:**

3: Good: Key resources (e.g., proofs, code, data) are available and key details (e.g., proofs, experimental setup) are sufficiently well-described for competent researchers to confidently reproduce the main results.

**Q3 Main Strengths:**

see below

**Q4 Main Weakness:**

see below

**Q5 Detailed Comments To The Authors:**

I'm afraid that, as told to the conference organizers, this paper is far from my
field of expertise and thus, all I could do is to verify that there are no typos
and that it is seemingly well written.

As a foreigner, though, the paper looks interesting and I assume it is novel,
but I could not assess on the novelty or importance of the methods employed
here. In the following, just a few comments are given with the hope they are
somehow useful to the authors and the other PC members:

First of all, I'm unsure of the importance that observations are provided by NNs
and, in general, the approach seems to be well-defined for any form of
probabilistic observations.

In page 4, the syntax \delta (s^k, \alpha^k) (s^{k+1}) looks strange and as far
as I can tell it was not introduced before. All notions, however, are properly
introduced and they are easy to read and understand.

A weak comment: in section 4 (page 5), the idea of "first finding an arbitrary
SPNE or SPCE and the iteratively freezing a set of variables and computing a new
SPNE or SPCE with an increasing social welfare" clearly resembles the idea of
progressing from local minima. Thus, the notion of convergence seems relevant
and should have been observed in the experimental sectino.

Finally, the discussion of the multi-agent case looks too short. Just the
computation of equilibria under these scenarios has some peculiarities that have
to be observed so I wonder whether they might affect the overal method described
here.



**Q7 Justification For Your Score:**

Just guessing, this paper falls really far from my field of expertise

**Q9 Complying With Reviewing Instructions:**

1: Yes.

---

### Official Review · Reviewer_1W1G · 2022-04-13

**Q2(1) Originality/Novelty:** 3
**Q2(2) Significance/Impact:** 3
**Q2(3) Correctness/Technical Quality:** 3
**Q2(6) Clarity Of Writing:** 3
**Q6 Overall Score:** 7
**Q8 Confidence In Your Score:** 3

**Q1 Summary And Contributions:**

The paper presents a  novel technique for neuro-symbolic concurrent stochastic games.

Formalizing the notion of equilibria for these models and focusing on the finite-horizon setting, and (global) social welfare subgame-perfect optimality, the paper presents two kinds pf equilibria: Nash equilibria and correlated equiibria.

**Q2 Assessment Of The Paper:**

More detailed information regarding each of these aspects is given below:

**Q2(4) Quality Of Experiments (Optional):**

3: Good: The experimental evaluation is adequate, and the results convincingly support the main claims.

**Q2(5) Reproducibility:**

3: Good: Key resources (e.g., proofs, code, data) are available and key details (e.g., proofs, experimental setup) are sufficiently well-described for competent researchers to confidently reproduce the main results.

**Q3 Main Strengths:**

The paper is relevant to AI.

The conceptual approach seems to be technically sound, with no obvious flaws in it.

The claims that the proposed algorithm is an improvement over existing ones is supported by theory and by experimental results.

The text is clear and well written. The paper is well organized, with well-structured sentences, where we can understand the development of the argument well. The abstract gives a good description of the paper, summarizing it well. Figures, diagrams and formulas are all readable and formatted correctly.


**Q4 Main Weakness:**

The research methodology could be improved – I do not think it is possible to reproduce the experiments using only the information in the paper (an important point in scientific research).

**Q5 Detailed Comments To The Authors:**

The approach to solve the proposed problem is novel, seems sound, and will have an impact in the field. It is well written, clear, and looks reproductible.

Therefore, the paper should be accepted.


**Q7 Justification For Your Score:**

This paper studies an interesting problem. It captures challenges in many practical applications. The proposed method seems reasonable. The proposal appears novel and could be useful for the AI community.

**Q9 Complying With Reviewing Instructions:**

1: Yes.

---

### Official Review · Reviewer_Cy15 · 2022-04-20

**Q2(1) Originality/Novelty:** 3
**Q2(2) Significance/Impact:** 3
**Q2(3) Correctness/Technical Quality:** 3
**Q2(6) Clarity Of Writing:** 3
**Q6 Overall Score:** 6
**Q8 Confidence In Your Score:** 4

**Q1 Summary And Contributions:**

The paper have considered finite-horizon equilibria computation for Concurrent Stochastic Games whose agents are equipped with NN-based perception mechanisms. It first shows that an exact solution based on backward induction may yield arbitrarily bad equilibria. Then it proposes an approximation algorithm called frozen subgame improvement. The experimentation is carried out on two case studies: an automated car-parking system and an aircraft collision avoidance system.

**Q2 Assessment Of The Paper:**

More detailed information regarding each of these aspects is given below:

**Q2(4) Quality Of Experiments (Optional):**

3: Good: The experimental evaluation is adequate, and the results convincingly support the main claims.

**Q2(5) Reproducibility:**

3: Good: Key resources (e.g., proofs, code, data) are available and key details (e.g., proofs, experimental setup) are sufficiently well-described for competent researchers to confidently reproduce the main results.

**Q3 Main Strengths:**

The paper is well written  and results are technically sound and different claims are well-supported by theoretical analysis and experimental results.

**Q4 Main Weakness:**

Lack of a theoretical complexity study and execution time in the experiments.

**Q5 Detailed Comments To The Authors:**

It would be interesting to add some details about the theoretical complexity and the  execution time in the experiments.

**Q7 Justification For Your Score:**

The results are technically sound and different claims are well-supported by theoretical analysis and experimental results.

**Q9 Complying With Reviewing Instructions:**

1: Yes.

---

### Decision · Program_Chairs · 2022-05-15

**Decision:**

Accept (Poster)

**Comment:**

Meta Review: This paper receives positive reviews from the reviewers. The major merit such as novelty and soundness are in particular appreciated by the reviewers. The experimental results strongly support the proposed methods in the paper.